# Hierarchical encoding of natural sound mixtures in ferret auditory cortex

**Agnès Landemard[1,2]\*, Célian Bimbard[2], Yves Boubenec[1]\***

[1]Laboratoire des systèmes perceptifs, Département d'études cognitives, École normale supérieure, PSL University, Paris, France; [2]UCL Institute of Ophthalmology, University College London, London, United Kingdom

### eLife Assessment

This paper presents **valuable** findings on the processing of sound mixtures in the auditory cortex of ferrets, a species widely used for studies of auditory processing. Using the convenient and relatively high-resolution method of functional ultrasound imaging, the authors provide **convincing** evidence that background noise invariance emerges across the auditory cortical processing hierarchy. They also draw informative comparisons with previously published fMRI data obtained in humans. This work will be of interest to researchers studying the auditory cortex and the neural mechanisms underlying auditory scene analysis and hearing in noise.

**\*For correspondence:**
agnes.landemard@hotmail.fr (AL);
boubenec@ens.fr (YB)

**Competing interest:** The authors declare that no competing interests exist.

**Abstract** Extracting relevant auditory signals from complex natural scenes is a fundamental challenge for the auditory system. Sounds from multiple sources overlap in time and frequency. In particular, dynamic 'foreground' sounds are often masked by more stationary 'background' sounds. Human auditory cortex exhibits a hierarchical organization where background-invariant representations are progressively enhanced along the processing stream, from primary to non-primary regions. However, we do not know whether this organizational principle is conserved across species and which neural mechanisms drive this invariance. To address these questions, we investigated background invariance in ferret auditory cortex using functional ultrasound imaging, which enables large-scale, high-resolution recordings of hemodynamic responses. We measured responses across primary, secondary, and tertiary auditory cortical regions as ferrets passively listened to mixtures of natural sounds and their components in isolation. We found a hierarchical gradient of background invariance, mirroring findings in humans: responses in primary auditory cortex reflected contributions from both foreground and background sounds, while background invariance increased in higher-order auditory regions. Using a spectrotemporal filter-bank model, we found that in ferrets this hierarchical structure could be largely explained by tuning to low-order acoustic features. However, this model failed to fully account for background invariance in human non-primary auditory cortex, suggesting that additional, higher-order mechanisms are crucial for background segregation in humans.

## Introduction

In the intricate auditory landscapes of our surroundings, the brain quickly extracts meaningful information from a cacophony of overlapping sources. Humans are remarkably good at detecting speech in complex noise: two people in a bar can easily have a conversation despite the ambient chatter. This ability is thought to rely on the different properties of background and foreground sounds, which are mixed in a single waveform at the level of the cochlea. Dynamic 'foreground' sounds, such as speech or vocalizations, fluctuate over short timescales, and these fluctuations convey new information over

time (*Singh and Theunissen, 2003*). In contrast, stationary 'background' sounds, like ambient chatter or the rustle of wind, fluctuate over longer periods, and are thus more predictable and less vital for signaling sudden events. They can be synthesized realistically based solely on their time-averaged acoustic statistics (*McWalter and McDermott, 2018*; *McDermott and Simoncelli, 2011*; *McWalter and Dau, 2017*), which the brain uses to perceptually fill short acoustic gaps (*McWalter and McDermott, 2019*). Despite these differences in the nature of foreground and background sounds, we still lack a general understanding of where background invariance occurs in the brain and what neural mechanisms underlie it.

In humans, multiple stages of processing build up background-invariant representations. In fMRI recordings of auditory cortex, non-primary areas are more background-invariant than primary areas (*Kell and McDermott, 2019*). In addition, intracranial cortical responses show rapid adaptation to new background noise, which improves representations of foreground speech (*Khalighinejad et al., 2019*). This effect is stronger for neurons in non-primary auditory cortex. This progressive invariance is unlikely to be fully explained by the parallel hierarchical organization of speech and music processing in humans (*Norman-Haignere et al., 2015*; *Norman-Haignere and McDermott, 2018*) since it is still present when excluding speech and music from foreground sounds (*Kell and McDermott, 2019*). Moreover, background invariance appears to be independent of selective attention (*Kell and McDermott, 2019*; *Khalighinejad et al., 2019*). Thus, the hierarchical organization of background invariance may not be specific to humans but rather a generic principle shared with other animals.

Background invariance can also be found in non-human animals. In fact, neurons that exhibit background-invariant responses are present throughout the auditory pathway in multiple species, increasingly from subcortical nuclei to the auditory cortex (*Souffi et al., 2020*; *Rabinowitz et al., 2013*). Background-invariant representations in auditory cortex are necessary for behavioral discrimination of natural sounds in noise (*Town et al., 2023*). Neurons with foreground-invariant responses are also found in the non-human auditory cortex (*Bar-Yosef and Nelken, 2007*; *Ni et al., 2017*; *Hamersky et al., 2025*), but a critical transformation of representations within the cortical hierarchy could specifically enhance foreground representations (*Schneider and Woolley, 2013*; *Saderi et al., 2020*; *Carruthers et al., 2015*). However, these studies differ by species, regions, state (awake vs. anesthetized), type of background noise (synthetic vs. natural), and signal-to-noise ratio. Thus, it remains unclear whether a cortical gradient in background invariance, as seen in humans, is present in other animals.

Mechanistically, background invariance could come from the simple tuning properties of neurons. Auditory cortical neurons are known to be tuned to frequency and spectrotemporal modulations (*Miller et al., 2002*; *Shamma, 2009*), which are acoustic features that differ between background and foreground sounds (*Elie and Theunissen, 2015*). In ferrets, this low-level spectrotemporal tuning can explain large-scale responses to natural sounds in both primary and non-primary regions of auditory cortex (*Landemard et al., 2021*). Thus, differences in background invariance across regions could result from regional differences in such low-order tuning (*Moore et al., 2013*; *Ivanov et al., 2022*). In humans, this tuning can explain responses in primary auditory cortex (*Santoro et al., 2014*), but not in non-primary fields (*Norman-Haignere and McDermott, 2018*). Consequently, the sensitivity of human non-primary cortex to acoustic high-order acoustic features may underlie the hierarchical gradient in background invariance. However, it remains untested whether background invariance in the auditory cortex relies on neural mechanisms that are conserved across species.

Here, we asked (1) whether background invariance is hierarchically organized in the auditory cortex of a non-human mammal, the ferret, and (2) whether hierarchical background invariance could be explained by tuning to low-order acoustic properties that differentiate foreground-like and background-like natural sounds. We defined foreground and background sounds based on an acoustic criterion of stationarity. We measured large-scale hemodynamic responses in ferrets' auditory cortex as they passively listened to isolated and combined foregrounds and backgrounds. Leveraging the spatial resolution and coverage of functional ultrasound imaging (fUSI), we showed that background invariance increased from primary to secondary and tertiary areas of auditory cortex. We could predict these effects using a model based on frequency and spectrotemporal modulations, suggesting that the organization of background invariance in the ferret brain can be derived from neural spectrotemporal tuning. Finally, we showed that the same model could not explain as well the patterns

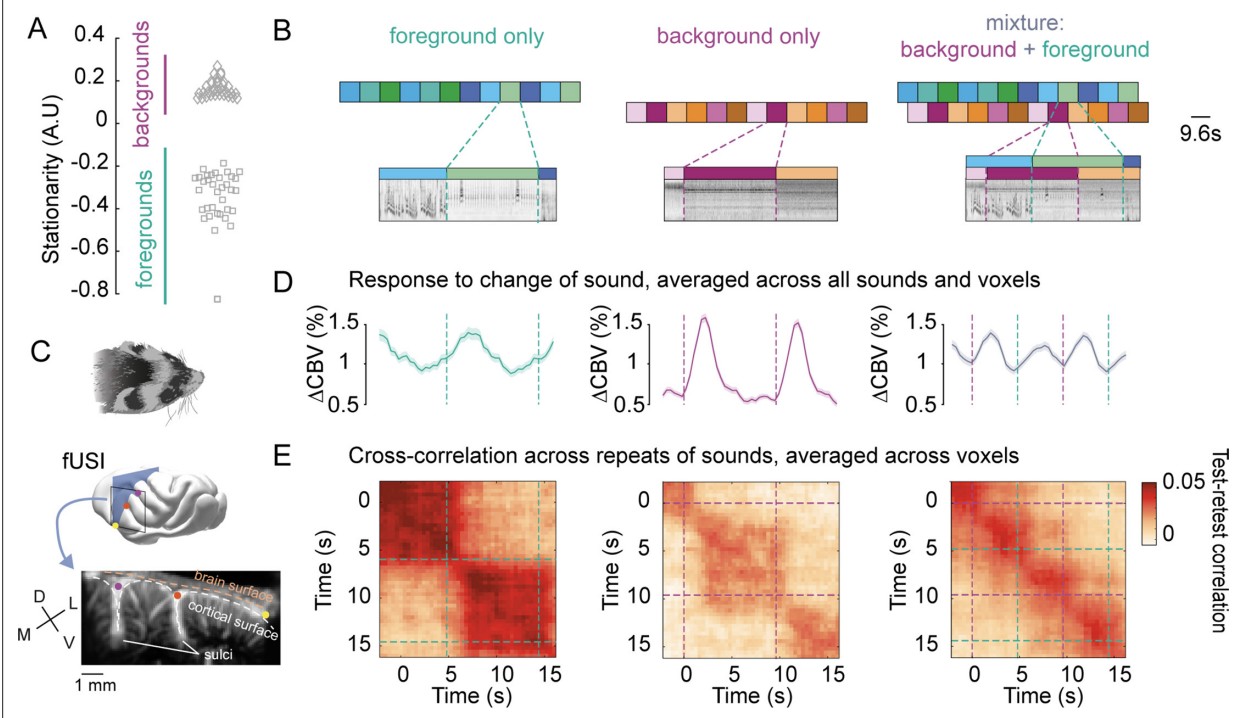

**Figure 1.** Hemodynamic activity reflects encoding of foregrounds and backgrounds. (**A**) Stationarity for foregrounds (*squares*) and backgrounds (*diamonds*). (**B**) Sound presentation paradigm, with example cochleagrams. We created continuous streams by concatenating 9.6 s foreground (*cold colors*) and background segments (*warm colors*) following the illustrated design. Each foreground (resp. background) stream was presented in isolation and with two different background (resp. foreground) streams. (**C**) We measured cerebral blood volume (CBV) in coronal slices (*blue plane*) of the ferret auditory cortex (*black outline*) with functional ultrasound imaging. We imaged the whole auditory cortex through successive slices across several days. Baseline blood volume for an example slice is shown, where two sulci are visible, as well as penetrating arterioles. *D*: dorsal, *V*: ventral, *M*: medial, *L*: lateral. (**D**) Changes in CBV aligned to sound changes, averaged across all (including non-responsive) voxels and all ferrets, as well as across all sounds within each condition (normalized to silent baseline). Shaded area represents standard error of the mean across sound segments. (**E**) Test-retest cross-correlation for each condition. Voxel responses for two repeats of sounds are correlated with different lags. Resulting matrices are then averaged across all responsive voxels (ΔCBV > 2.5%).

The online version of this article includes the following source data for figure 1:

**Source data 1.** List of sounds used in ferret experiments.

of background invariance found in human auditory cortex, suggesting that additional mechanisms underlie background invariance in humans.

## Results

Using fUSI, we measured hemodynamic responses in the auditory cortex of ferrets passively listening to continuous streams of natural sounds. We used three types of stimuli: foregrounds, backgrounds, and combinations of those. We use those terms to refer to sounds differing in their stationarity, under the assumption that stationary sounds carry less information than non-stationary sounds, and are thus typically ignored. To classify natural sounds into these categories, we computed each sound's stationarity, defined as the stability in time of a sound's acoustic statistics (*McWalter and McDermott, 2018*). Backgrounds were chosen as the most stationary and foregrounds as the least stationary sounds (*Figure 1A*, *Figure 1—source data 1*). Different sound segments were concatenated to form continuous streams (*Figure 1B*). In the isolated condition, a sound change occurred every 9.6 s. In the mixture condition, the same sequences overlapped with a lag of 4.8 s between the foreground and background streams, leading to a change in one or the other stream every 4.8 s.

Hemodynamic responses stably encoded sound identity after transient responses. We continuously measured hemodynamic activity in ferret auditory cortex (*Figure 1C*) throughout the sound sequences and normalized cerebral blood volume (CBV) relative to blocks of silent baseline. Sound changes

elicited transient increases in activity before reaching a sustained level (*Figure 1D*). Sustained activity was lower for backgrounds in isolation than for foregrounds and mixtures. We asked whether sound encoding was stable over time despite these changes in amplitude. We focused on the sound-specific activity of responsive voxels (ΔCBV > 2.5%) by subtracting their average timecourse across all sounds in each category. For each voxel, we cross-correlated vectors of responses to sounds of each category across two repetitions (*Figure 1E*). Voxels stably encoded sounds of each category throughout their duration despite the observed variation in response amplitude. Thus, we summarized voxels' activity by their time-averaged activity over an interval spanning 2.4–4.8 s after sound change.

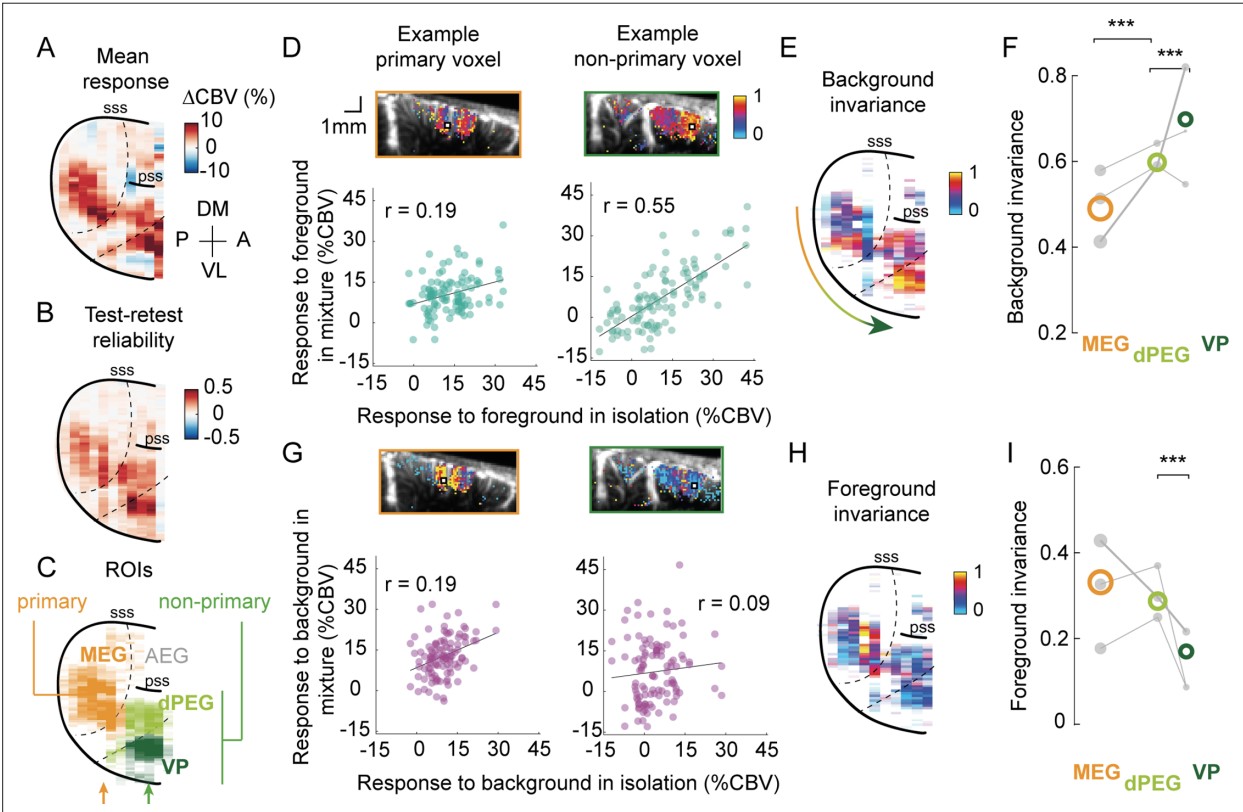

**Figure 2.** Invariance to background sounds is hierarchically organized in ferret auditory cortex. (**A**) Map of average response for an example hemisphere (ferret L). Responses are expressed in percent changes in cerebral blood volume (CBV) relative to baseline activity, measured in periods of silence. Values are averaged across depth to obtain this surface view of auditory cortex. (**B**) Map of test-retest reliability. In the following maps, only reliably responding voxels are displayed (test–retest > 0.3 for at least one category of sounds) and the transparency of surface bins in the maps is determined by the number of (reliable) voxels included in the average. (**C**) Map of considered regions of interest (ROIs), based on anatomical landmarks. The arrows indicate the example slices shown in (**D**) (*orange*: primary; *green*: non-primary example). (**D**) Responses to isolated and combined foregrounds. *Bottom*: responses to mixtures and foregrounds in isolation, for example, voxels (*left*: primary; *right*: non-primary). Each dot represents the voxel's time-averaged response to every foreground (*x-axis*) and mixture (*y-axis*), averaged across two repetitions. r indicates the value of the Pearson correlation. *Top*: maps show invariance, defined as noise-corrected correlation between mixtures and foregrounds in isolation, for the example voxel's slice with values overlaid on anatomical images representing baseline CBV. Example voxels are shown with white squares. (**E**) Map of background invariance for the same hemisphere (see *Figure 2—figure supplement 2* for other ferrets). (**F**) Quantification of background invariance for each ROI. Colored circles indicate median values across all voxels of each ROI, across animals. Gray dots represent median values across the voxels of each ROI for each animal. The size of each dot is proportional to the number of voxels across which the median is taken. The thicker line corresponds to the example ferret L. ***: $p \leq 0.001$ for comparing the average background invariance across animals for pairs of ROIs, obtained by a permutation test of voxel ROI labels within each animal. (**G–I**) Same as (**D–F**) for foreground invariance (comparing mixtures to backgrounds in isolation). *AEG*, anterior ectosylvian gyrus; *MEG*, medial ectosylvian gyrus; *dPEG*, dorsal posterior ectosylvian gyrus; *VP*, ventral posterior auditory field.

The online version of this article includes the following source data and figure supplement(s) for figure 2:

**Source data 1.** Table of statistics for comparison across regions.

**Figure supplement 1.** Invariance dynamics.

**Figure supplement 2.** Maps for all ferrets.

# Invariance to background sounds is hierarchically organized in ferret auditory cortex

We investigated the spatial organization of sound responses throughout the whole ferret auditory cortex. We found that reliable sound-evoked responses were confined to the central part of the ventral gyrus of the auditory cortex (*Figure 2A and B*). Hereafter, we focus on reliable sound-responsive voxels (test–retest correlation > 0.3 for sounds in at least one condition). We assigned voxels to three regions of interest in ferret auditory cortex (*Figure 2C*): MEG (primary), dPEG (secondary), and VP (tertiary, see *Elgueda et al., 2019*; *Landemard et al., 2021*; *Radtke-Schuller, 2018*).

Invariance to background sounds was stronger in non-primary than in primary auditory cortex. To probe invariance to background sounds for each voxel, we compared responses to mixtures and isolated foregrounds across sound segments (*Figure 2D*). Responses to mixtures and isolated foregrounds tended to be more similar for voxels in non-primary regions. We quantified this effect by computing a noise-corrected version of the Pearson correlation between responses to mixtures and foregrounds for each voxel, which we call background invariance (*Kell and McDermott, 2019*). Background invariance emerged shortly after a sound change and was stable throughout the duration of each sound snippet (*Figure 2—figure supplement 1*). When going up the hierarchy, correlations between mixtures and foregrounds increased (tertiary (VP) > secondary (dPEG) > primary (MEG), $p = 0.001$ with a permutation test), indicating a stronger invariance to background sounds in higher-order areas (*Figure 2E and F*, *Figure 2—figure supplement 2*). While background invariance was overall highest in VP, the differences within non-primary areas were more variable across animals (see *Figure 2—source data 1*).

To test whether foregrounds were processed differently than backgrounds, and at which stage, we also compared responses to mixtures and backgrounds in isolation. We defined a 'foreground invariance', symmetric to the background invariance, by correlating responses to mixtures and isolated backgrounds across sound segments (*Figure 2G*). In primary auditory cortex (MEG), foreground invariance was slightly lower than background invariance, although this difference was not significant ($p = 0.063$, obtained by randomly permuting the sounds' background and foreground labels, 1000 times). However, foreground invariance tended to decrease from primary to non-primary areas (*Figure 2H and I*, VP < dPEG: $p = 0.001$, dPEG < MEG: $p = 0.136$ with a permutation test). As a consequence, foreground invariance became significantly lower than background invariance in both dPEG ($p = 0.001$) and VP ($p = 0.001$).

## A model of auditory processing predicts hierarchical differences

By definition, foregrounds and backgrounds are separated in the acoustic domain. Low-level acoustic features computed from their waveforms can well discriminate both categories. To show this, we used a standard two-stage filter-bank model of auditory processing (*Chi et al., 2005*). This model (1) computes a sound's cochleagram, (2) convolves the resulting cochleagram through a bank of spectrotemporal modulation filters, tuned to a range of frequencies and spectrotemporal modulations (*Figure 3A*). Foregrounds and backgrounds differed in their pattern of energy in the modulation space (*Figure 3B and C*). In particular, the axis of temporal modulation was highly discriminative: foregrounds tended to have higher energy in the low rates (< 8 Hz), and backgrounds in higher rates (> 8 Hz).

In ferret auditory cortex, large-scale cortical responses to natural sounds are well predicted by the spectrotemporal model, even when controlling for stimuli correlations (*Landemard et al., 2021*). To test whether differences in neural invariance could be explained by tuning to low-order acoustic features, we predicted the voxels' responses using this model. We derived voxels weights for model features using cross-validated ridge regression of trial-averaged responses to all sounds (*Figure 3D and E*).

Tuning to frequency and spectrotemporal modulations showed large-scale spatial structure. Through the model weights, we first explored whether tuning to acoustic features was topographically organized. We extracted the voxels' preferred frequency, spectral and temporal modulations by marginalizing over the other features. We retrieved the tonotopic organization of the auditory cortex, confirming our anatomical segmentation into functional areas (*Elgueda et al., 2019*). The voxels' preferred spectral and temporal modulations were also spatially organized (*Figure 3F*, *Figure 3—figure supplement 1* for all ferrets). To directly examine the differences of tuning between non-primary

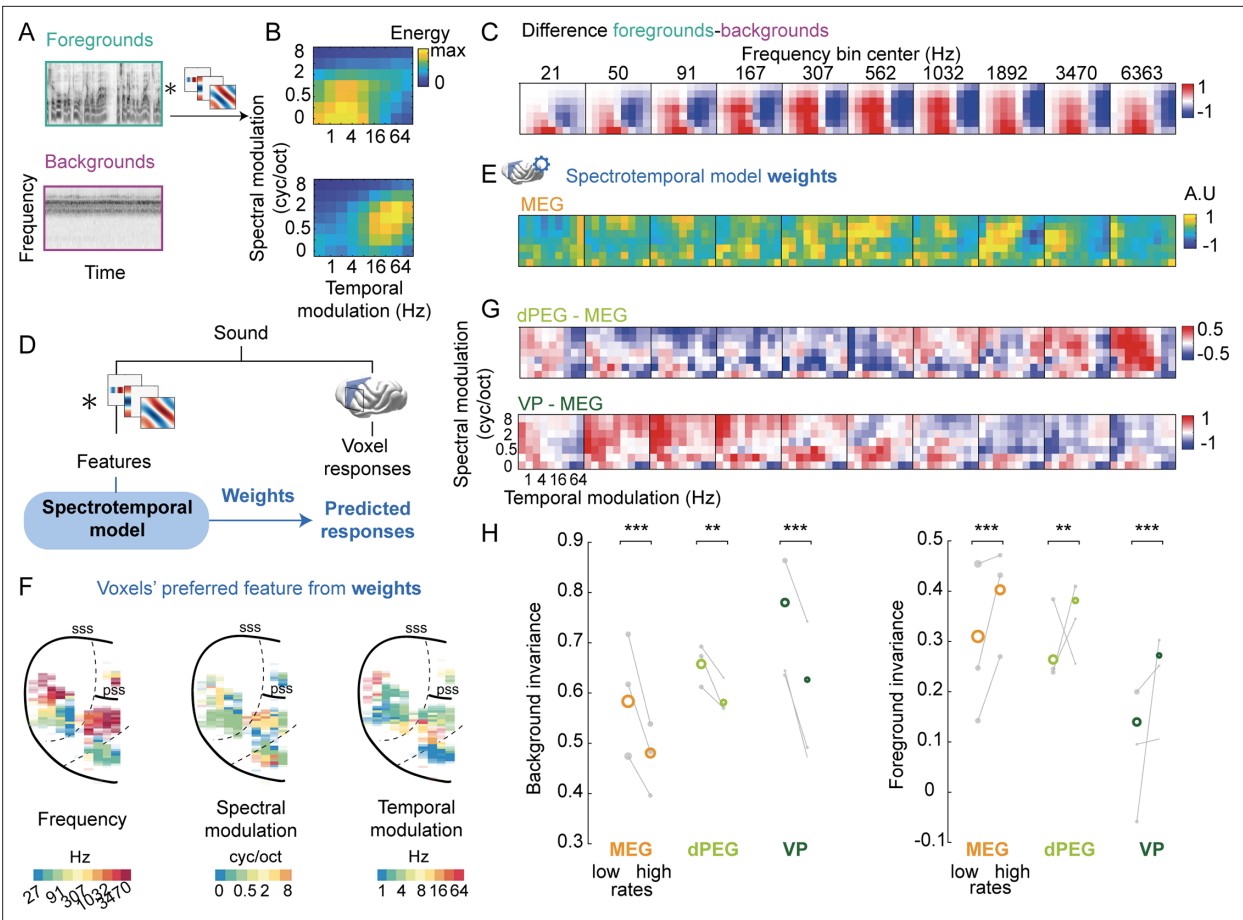

**Figure 3.** Simple spectrotemporal tuning explains spatial organization of background invariance. (**A**) Presentation of the two-stage filter-bank, or spectrotemporal model. Cochleagrams (shown for an example foreground and background) are convolved through a bank of spectrotemporal modulation filters. (**B**) Energy of foregrounds and backgrounds in spectrotemporal modulation space, averaged across all frequency bins. (**C**) Average difference of energy between foregrounds and backgrounds in the full acoustic feature space (frequency * temporal modulation * spectral modulation). (**D**) We predicted time-averaged voxel responses using sound features derived from the spectrotemporal model presented in (**A**) with ridge regression. For each voxel, we thus obtain a set of weights for frequency and spectrotemporal modulation features, as well as cross-validated predicted responses to all sounds. (**E**) Average model weights for MEG. (**F**) Maps of preferred frequency, temporal and spectral modulation based on the fit model. To calculate the preferred value for each feature, we marginalized the weight matrix over the two other dimensions. (**G**) Average differences of weights between voxels of each non-primary (dPEG and VP) and primary (MEG) region. (**H**) Background invariance (*left*) and foreground invariance (*right*) for voxels tuned to low (< 8 Hz) or high (> 8 Hz) temporal modulation rates within each region of interest (ROI). Colored circles indicate median value across all voxels of each ROI, across animals. Gray dots represent median values across the voxels of each ROI for each animal. **: $p \leq 0.01$, ***: $p \leq 0.001$ for comparing the average background invariance across animals for voxels tuned to low vs. high rates, obtained by a permutation test of tuning within each animal.

The online version of this article includes the following figure supplement(s) for figure 3:

**Figure supplement 1.** Tuning to acoustic features for all ferrets.

and primary areas, we computed the differences in weights between dPEG vs. MEG and VP vs. MEG, throughout the whole three-dimensional acoustic space (*Figure 3G*). The weights of voxels in non-primary areas differed from those in primary areas in a complex manner, in contrast to the simple separation of the acoustic space between foregrounds and backgrounds (*Figure 3C*). Compared to MEG, dPEG was preferentially tuned to high frequencies and higher spectral modulations (*Figure 3G*, top), while VP tuning was biased toward low frequencies and intermediate spectrotemporal modulations (*Figure 3G*, bottom).

We sought to assess whether these differences in tuning underlaid differences in background invariance across regions. The model included a range of realistic temporal rates, and this axis was the most informative to discriminate foregrounds from backgrounds (*Figure 3C*). Thus, we first compared the

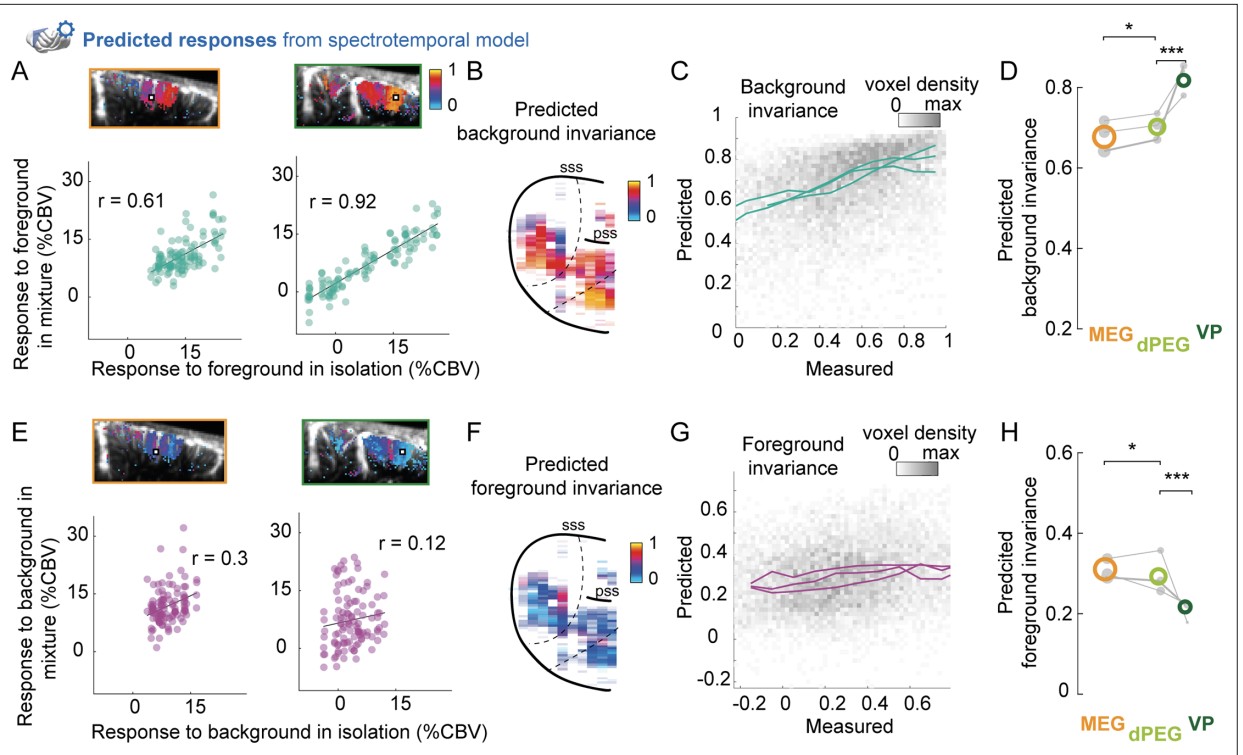

**Figure 4.** A model of auditory processing predicts hierarchical differences in ferret auditory cortex. Same as in *Figure 2* using cross-validated predictions from the spectrotemporal model. (**A**) Predicted responses to mixtures and foregrounds in isolation for example voxels (*left*: primary; *right*: non-primary). Each dot represents the voxel's predicted response to foregrounds (*x-axis*) and mixtures (*y-axis*). r indicates the value of the Pearson correlation. Maps above show predicted invariance values for the example voxel's slice overlaid on anatomical images representing baseline cerebral blood volume (CBV). Example voxels are shown with white squares. (**B**) Maps of predicted background invariance, defined as the correlation between predicted responses to mixtures and foregrounds in isolation. (**C**) Binned scatter plot representing predicted vs. measured background invariance across voxels. Each line corresponds to the median across voxels for one animal, using 0.1 bins of measured invariance. (**D**) Predicted background invariance for each region of interest (ROI). Colored circles indicate median value across all voxels of each ROI, across animals. Gray dots represent median values across the voxels of each ROI, for each animal. The size of each dot is proportional to the number of voxels across which the median is done. The thicker line corresponds to example ferret L. *: $p \leq 0.05$; ***: $p \leq 0.001$ for comparing the average predicted background invariance across animals for pairs of ROIs, obtained by a permutation test of voxel ROI labels within each animal. (**E–H**) Same as (**A–D**) for predicted foreground invariance, that is, comparing predicted responses to mixtures and backgrounds in isolation.

The online version of this article includes the following figure supplement(s) for figure 4:

**Figure supplement 1.** Assessment and effect of model prediction accuracy across species.

**Figure supplement 2.** Predicting from a model fitted on isolated sounds only.

invariance properties of voxels tuned to low or high temporal modulation rates. Within each region, voxels tuned to lower modulation rates displayed higher background invariance and lower foreground invariance than those tuned to higher rates (*Figure 3H*, $p < 0.004$ within each region of interest (ROI) by permuting voxels' tuning, 1000 times). However, within each tuning group, differences were still present across regions (primary vs. non-primary, $p = 0.001$ within voxels tuned to high rates and within voxels tuned to low rates, see *Figure 2—source data 1* for more detailed comparisons between regions), confirming that tuning to temporal modulation was not the only mechanism at play.

To take into account tuning across all dimensions, we then directly looked at the invariance of responses predicted from the spectrotemporal model. The model was able to predict voxel responses accurately (*Figure 4—figure supplement 1*) and recapitulated invariance properties across single voxels (*Figure 4A–C and E–G*). To verify how well the model could explain the observed invariance, we directly compared measured and predicted invariance across voxels for each animal, regardless of ROI (*Figure 4C and G*). Predictions explained a large fraction of the reliable variability in background invariance (mean noise-corrected correlation between measured and predicted invariance: $r = 0.65 \pm 0.007$ standard error of the mean across animals) and foreground invariance (mean

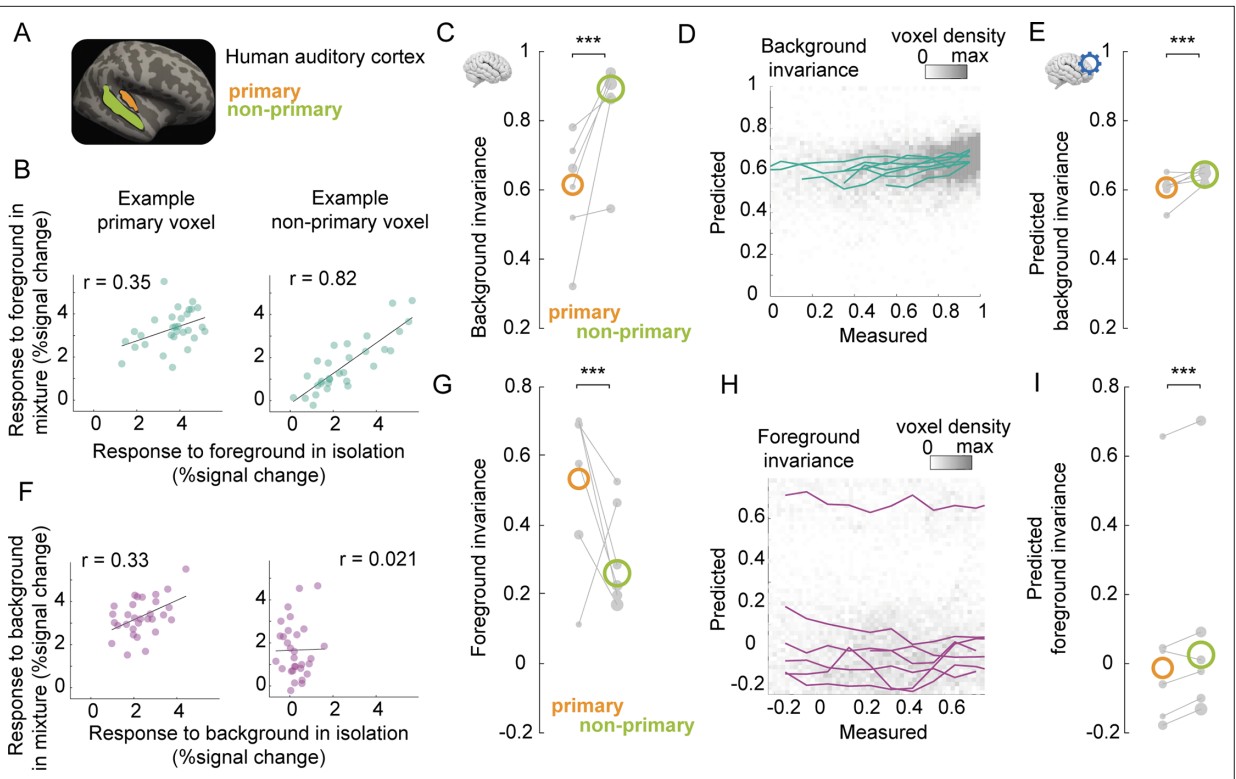

**Figure 5.** The spectrotemporal model is a poor predictor of human background invariance. (A) We replicated our analyses with a dataset of a similar experiment measuring fMRI responses in human auditory cortex (*Kell and McDermott, 2019*). We compared responses in primary and non-primary auditory cortex, as delineated in *Kell and McDermott, 2019*. (B) Responses to mixtures and foregrounds in isolation for example voxels (*left*: primary; *right*: non-primary). Each dot represents the voxel's response to foregrounds (*x-axis*) and mixtures (*y-axis*), averaged across repetitions. r indicates the value of the Pearson correlation. (C) Quantification of background invariance measured for each region of interest (ROI). Colored circles indicate median value across all voxels of each ROI, across subjects. Gray dots represent median values for each ROI and subject. The size of each dot is proportional to the number of (reliable) voxels across which the median is done. *: $p \leq 0.05$; ***: $p \leq 0.001$ for comparing the average predicted background invariance across subjects for pairs of ROIs, obtained by a permutation test of voxel ROI labels within each subject. (D) Binned scatter plot representing predicted vs measured background invariance across voxels. Each line corresponds to the median across voxels for one subject, using 0.1 bins of measured invariance. (D) Same as (C) for responses predicted from the spectrotemporal model. (F–I) Same as (B–E) for foreground invariance, that is, comparing predicted responses to mixtures and backgrounds in isolation.

The online version of this article includes the following figure supplement(s) for figure 5:

**Figure supplement 1.** Spectrotemporal tuning properties for humans.

**Figure supplement 2.** Invariance metrics are not affected by differences in test–retest reliability across regions.

$r = 0.43 \pm 0.04$). Predicted responses recapitulated the gradient from primary to non-primary areas for background invariance (*Figure 4D*; MEG < dPEG, $p = 0.022$ ; dPEG < VP, $p = 0.001$, permutation test) and foreground invariance (*Figure 4H*; MEG > dPEG: $p = 0.014$, dPEG > VP: $p = 0.001$, permutation test). Results were similar if the model was fit solely on isolated sounds, excluding mixtures from the training set (*Figure 4—figure supplement 2*).

Thus, hierarchical differences in invariance were largely explained by differences in tuning to frequency and spectrotemporal modulation between regions.

## Species difference in background invariance

In ferrets, background invariance is hierarchically organized, and this organization can largely be explained by simple spectrotemporal tuning. Is it also the case for the humans' hierarchical organization of invariance? To test this, we compared background invariance previously obtained in humans with what we found in ferrets. We used a comparable published dataset (*Kell and McDermott, 2019*), in which combinations of foregrounds and backgrounds were presented to human subjects, in isolation and in mixtures. The BOLD signal from the auditory cortex was acquired with fMRI (*Figure 5A*).

Applying our analyses to this dataset, we reproduced the results of *Kell and McDermott, 2019* and showed that background invariance was stronger in non-primary than in primary auditory cortex (*Figure 5B and C*, non-primary > primary, $p = 0.001$, permutation test on the average across seven subjects). In parallel, we found that foreground invariance decreased from primary to non-primary areas (*Figure 5F and G*, non-primary < primary, $p = 0.001$, permutation test). As in ferrets, foreground and background invariance values were similar in primary auditory cortex ($p = 0.577$, permutation test), but significantly differed in non-primary auditory cortex ($p = 0.002$, permutation test).

We next tested whether the spectrotemporal model could predict human voxel responses. The model performed similarly in humans and ferrets (*Figure 5—figure supplement 1*, *Figure 4—figure supplement 1*, $r = 0.42 \pm 0.01$ standard error of the mean across subjects, humans vs ferrets: $p = 0.0667$, Wilcoxon rank-sum test). However, the model explained less of the measured background invariance in humans than in ferrets (*Figure 5D*, noise-corrected correlation between measured and predicted invariance: $r = 0.23 \pm 0.04$, $p = 0.0167$ compared to ferrets, $p = 0.0156$ compared to 0, Wilcoxon rank-sum test). While predicted background invariance differed significantly between primary and non-primary regions (*Figure 5E*, primary < non-primary, $p = 0.001$, permutation test), the predicted difference was much smaller than observed. Furthermore, the model failed to predict the levels of foreground invariance in both primary and non-primary human auditory cortex (*Figure 5H and I*, noise-corrected correlation between measured and predicted invariance: $r = -0.17 \pm 0.06$, $p = 0.0167$ compared to ferrets, $p = 0.0312$ compared to 0, Wilcoxon rank-sum test). Thus, the spectrotemporal model performed worse in explaining the patterns of cortical invariance in humans than in ferrets. This suggests that additional higher-order mechanisms contribute to background segregation in humans.

## Discussion

In this work, we investigated large-scale background and foreground invariance in auditory cortex using mixtures of natural sounds. In ferrets, we found a hierarchical gradient of invariance, where primary cortical areas encode both background and foreground sounds, while non-primary regions show increasing invariance to background sounds. A spectrotemporal filter-bank model could account for a large portion of the hierarchical changes in ferrets, suggesting that low-level tuning to frequency and modulation properties explain how stationary 'background' and more dynamic 'foreground' signals are segregated in hemodynamic signals. When we applied the same model to human data, we found that although humans also display a hierarchical increase in background invariance, the model failed to capture a substantial fraction of the observed effect in non-primary regions. This suggests that mechanisms beyond simple spectrotemporal tuning are at play in human auditory cortex.

Our interpretation of foreground and background neural representations relies on several methodological choices that make the problem tractable but introduce specific limitations. First, this study defined foregrounds and backgrounds solely based on their acoustic stationarity, rather than perceptual judgments. This choice allowed us to isolate the contribution of acoustic factors in a simplified setting. Within this controlled framework, we show that acoustic features of foreground and background sounds drive their separation in the brain and the hierarchical extraction of foreground sound features. Second, we used blood volume as a proxy for local neuronal activity. Thus, our signal ignores any heterogeneity that might exist at the level of local neuronal populations. However, our main findings are related to the large-scale organization of cortical responses and how they relate to those of humans. For this purpose, the functional spatial resolution of our signal, driven by the spatial resolution of neurovascular coupling, should be adapted. In addition, using hemodynamic signals provides a much better comparison with human fMRI data, where the same limitations are present.

Our results show that foregrounds and backgrounds are similarly represented in primary auditory cortex, both in ferrets and humans; at this stage, we found no significant bias toward foregrounds. This is in line with previous work in non-human animals, including birds, where neurons in various regions up to primary auditory cortex display a continuum of background-invariant and foreground-invariant responses (*Ni et al., 2017*; *Bar-Yosef and Nelken, 2007*; *Schneider and Woolley, 2013*). In ferrets, primary auditory cortex has been found to over-represent backgrounds in mixtures compared to foregrounds (*Hamersky et al., 2025*). In contrast, we found a slight, non-significant bias toward foregrounds in primary regions. This difference could be driven by a difference in timescales, as we looked at slower timescales in which adaptation might be more present, reducing the strength of background

encoding. In humans, we found a much smaller gap between background and foreground invariance in primary auditory cortex, which was not predicted by the spectrotemporal model. Additionally, more closely controlled experiments would be needed to confirm and understand this species difference.

The encoding of background in primary regions could play a functional role in background denoising at later stages of processing. Studies show complex synergetic interactions between target sounds and background noise in primary auditory cortex, even with simple stimuli. On the one hand, white noise can sharpen neurons' tuning curves, enhancing tone discrimination (*Christensen et al., 2019*), and improve the representations of frequency sweeps (*Malone et al., 2017*). On the other hand, background information can be more easily decoded from population responses when presented with a target sound than when in isolation (*Malone et al., 2017*). Computationally, background-related information could facilitate further extraction of foregrounds. For instance, in a convolutional neural network trained to visually recognize digits in different types of noise, when local feedback is implemented, early layers encode noise properties, while later layers represent clean signals (*Lindsay et al., 2022*). Similarly, primary auditory areas could be recruited to maintain background representations, enabling downstream cortical regions to use these representations to specifically suppress background information and enhance foreground representations (*Hicks and McDermott, 2024*).

We found that the main difference regarding background invariance between humans and ferrets was in how well invariance could be explained by a model based on low-level acoustic features. There could be multiple reasons for this difference.

First, methodological differences could contribute to the observed species difference. In our ferret experiment, sounds were presented continuously and in longer segments, an experimental design made possible by fUSI but impractical for fMRI due to scanner noise. However, this is unlikely to have a major impact, as our analyses focused on the sustained response component, similar to the fMRI experiment. Furthermore, while fUSI and fMRI both provide hemodynamic signals, they differ in their physiological basis (CBV vs. BOLD). Studies comparing CBV-weighted and BOLD fMRI indicate that CBV provides a more spatially and temporally precise signal than BOLD (*Silva et al., 2007*), improving localization of neural activity. In addition, fMRI has a worse spatial resolution than fUSI (here, 2 vs. 0.1 mm voxels). However, this difference in resolution compensates for the difference in brain size between humans and ferrets. In our previous work, we showed that a large fraction of cortical responses to natural sounds could be predicted from one species to the other using these methods (*Landemard et al., 2021*). Last, to minimize potential biases, we applied the same criteria for voxel inclusion and noise-correction procedures across datasets. Thus, while minor differences in data quality and experimental protocols exist, they are unlikely to fully account for the species differences we observed.

Second, ferret and human paradigms differed by their control of attention. Ferrets were passively listening to the sounds while human participants had to perform a simple loudness-based task to ensure attention to the sounds. Even though specific auditory attention is not necessary for the gradient in invariance (*Kell and McDermott, 2019*), the task ensured that subjects remained engaged, while this was not the case for ferrets. General engagement might still affect auditory representations via top-down processes (*Elhilali et al., 2009*; *Saderi et al., 2021*; *Schwartz and David, 2018*). Thus, we cannot exclude that both species possess a similar feedforward acoustic analysis (well modeled by simple spectrotemporal filters) as well as more context-driven networks in non-primary auditory cortex, which were more strongly engaged in humans than in ferrets due to experimental procedures. In addition, most of the sounds included in our study likely have more relevance for humans compared to ferrets (see *Figure 1—source data 1*). Despite including ferret vocalizations and environmental sounds that are more ecologically relevant for ferrets, it is not clear whether ferrets would behaviorally categorize foregrounds and backgrounds as humans do. Examining how ferrets naturally orient or respond to foreground and background sounds under more ecologically valid conditions, potentially with free exploration or spontaneous listening paradigms, could help address this issue.

Third, human non-primary regions could host higher-order computations that cannot be predicted by simple acoustic features and are not present in ferrets. Human and non-human primates have a complex functional architecture in the auditory belt and parabelt regions (*Hackett, 2010*), likely supporting elaborate computations. These computations might be related to conspecific calls and required for advanced vocal communication. In particular, the unique importance of speech for humans has shaped auditory circuits throughout the brain. Neural circuits in human non-primary

auditory cortex encode higher-order acoustic features, such as phonemic patterns, which might help to segregate meaningful content from background noise (*Norman-Haignere et al., 2015*; *Mesgarani et al., 2014*). Last, other processes might critically shape background-invariant representations in humans, such as dynamic adaptation to backgrounds that could be faster in humans than in ferrets (*Khalighinejad et al., 2019*).

The divergence in background invariance mechanisms between humans and ferrets may reflect differences in the depth and complexity of their auditory cortical hierarchies. Our previous work (*Landemard et al., 2021*) showed that large-scale representations of natural sounds are not clearly distinct between primary and non-primary auditory regions in the ferret cortex. However, *Sabat et al., 2025* found a hierarchical organization of temporal integration in ferrets, reminiscent of humans (*Norman-Haignere et al., 2022*). In the present study, we found a gradient of background invariance that was similarly structured as in humans, yet relying on different mechanisms. Altogether, this raises a fundamental question: Does the ferret auditory cortex correspond to the different early auditory fields in the human core regions, with deeper hierarchical stages simply being absent? Or do these differences reflect evolutionary divergence, with partly shared computational principles and homologous hierarchy, but additional cortical modules in humans, enabling for instance more sophisticated background segregation or dedicated speech and music processing (*Norman-Haignere and McDermott, 2018*)? Addressing these questions will require a finer characterization of cross-species homologies, leveraging high-resolution functional imaging and electrophysiological recordings to uncover whether hierarchical transformations in auditory processing follow a conserved or species-specific trajectory.

## Materials and methods
### Animal preparation
Experiments were performed in three female ferrets (B, L, and R), across one or both hemispheres (left for all ferrets + right for R). Experiments were approved by the French Ministry of Agriculture (protocol authorization: 21022) and strictly comply with the European directives on the protection of animals used for scientific purposes (2010/63/EU). Animal preparation and fUS imaging were performed as in *Bimbard et al., 2018*. Briefly, a metal headpost was surgically implanted on the skull under anesthesia. After recovery from surgery, a craniotomy was performed over auditory cortex and then sealed with an ultrasound-transparent polymethylpentene (TPX) cover, embedded in an implant of dental cement. Animals could then recover for 1 week, with unrestricted access to food, water, and environmental enrichment.

fUSI data were collected using an Iconeus One system (Iconeus, Paris, France) and a linear IcoPrime ultrasonic probe (15 MHz central frequency, 70% bandwidth, 0.110 mm pitch, 128 elements). Complex frames are acquired at 500 Hz. To remove tissue motion, a spatio-temporal clutter filter is applied using sets of 200 contiguous frames (*Demené et al., 2015*). The power Doppler signal is then obtained by taking the mean power over these 200 frames, thus getting an effective 2.5 Hz sampling rate. The power Doppler signal is approximately proportional to CBV (*Macé et al., 2011*).

In each recording day, we imaged activity in response to the entire sound set sequentially in two coronal slices. The probe was angled at ~30–35° relative to the vertical axis in order for it to be roughly parallel to the surface of the brain. Through several recording sessions, we covered primary to tertiary regions of auditory cortex. Coronal slices were 14 mm wide and spaced 0.4 mm apart. The resolution of each voxel was 0.1 * 0.1 * ~0.4 mm (the latter dimension, called elevation, is dependent on the depth of the voxel).

### Sound presentation
Ferrets were awake, head-fixed, and passively listening to different streams of natural sounds: foregrounds in isolation, backgrounds in isolation, or mixtures of both.

### Sound stimuli
To choose foreground vs. background sounds, we used a previously established method based on sounds' stationarity (*Kell and McDermott, 2019*; *McWalter and McDermott, 2019*). We reproduce here a description of this algorithm from *McWalter and McDermott, 2019*.

To quantify the stationarity of real-world sound recordings, we computed the standard deviation of the texture statistics measured across successive segments of a signal for a variety of segment lengths (0.125, 0.25, 0.5, 1, and 2 s). For each segment length, we performed the following steps. First, we divided a signal into adjacent segments of the specified duration and measured the statistics in each segment. Second, we computed the standard deviation across the segments. Third, we averaged the standard deviation across statistics within each statistic class (envelope mean, envelope coefficient of variation, envelope skewness, envelope correlations, modulation power, and modulation correlations). Fourth, we normalized the average standard deviation for a class by its median value across sounds (to put the statistic classes on comparable scales, and to compensate for any differences between statistics in intrinsic variability). Fifth, we averaged the normalized values across statistics classes. Sixth, we averaged the resulting measure for the five segment lengths to yield a single measure of variability for each real-world sound recording. Finally, we took the negative logarithm of this quantity, yielding the stationarity measure.

We then defined arbitrary thresholds to define foregrounds and backgrounds, so that both groups were well separated in the stationarity axis (*Figure 1A*). Sounds were taken from *Kell and McDermott, 2019* and *Norman-Haignere et al., 2015* or downloaded from online resources (see *Figure 1—source data 1* for the full list of sounds).

## Sound design and presentation

Foreground and background streams were composed of sequences of six different sound snippets of 9.6 s duration, repeated twice, so that the same snippet would occur at two distinct times of the stream (*Figure 1B*). For each condition, six such streams were created by drawing randomly from a pool of 36 individual snippets. Foreground and background streams were presented in isolation and in mixtures. In mixtures, both streams were overlaid with a delay of 4.8 s so that a change, either of foreground or background, occurred every 4.8 s. Each foreground stream was presented with two different background streams, and vice versa. In this design, each foreground snippet was presented with four different backgrounds, and in two history contexts.

Recording sessions were composed of three 'runs' in which two foreground streams and two background streams were presented in all possible combinations, each preceded by 20 s of silence to establish baseline activity. These runs were presented twice. The order of sounds within each run was fixed, but the relative order of runs was randomized for each recording session. A list of sounds used is available in *Figure 1—source data 1*.

Sounds were presented at 65 dB SPL. In mixtures, foregrounds and backgrounds were presented with a similar gain (0 dB SNR). Each snippet was normalized by its root mean square prior to integration in the sequence.

## Data analysis

### Denoising and normalization

To correct for large movements of the brain or deformations throughout the session, we used NoRMcorre, a non-rigid motion correction algorithm developed for two-photon imaging (*Pnevmatikakis and Giovannucci, 2017*).

Data were then denoised with canonical correlation analysis (CCA) to remove components shared between regions inside and outside of the brain, as described in *Landemard et al., 2021* with the difference that data was not recentered beforehand.

We normalized the signal in each voxel by subtracting and then dividing by average baseline activity. Baseline activity was estimated during 20 s segments of silence occurring every ~20 min, before each run. The obtained value was called $\Delta CBV$ and expressed in percent changes in CBV.

### Combining different recordings

The whole dataset consisted of multiple slices, each recorded in a different recording session. Slices to image on a given day were chosen at random to avoid any systematic bias. Responses were consistent across neighboring slices recorded on different sessions, as shown by the maps of average responses (*Figure 2A*, *Figure 2—figure supplement 2*) where any spatial continuity across different slices must reflect a true underlying signal in the absence of common noise.

## Cross-correlation matrices

For each voxel, we computed the Pearson correlation between the vectors of responses to sounds of a given category (either foregrounds, backgrounds, or mixtures) for two repeats, and with different lags. We then averaged these matrices across all voxels to obtain the cross-correlation matrices shown in *Figure 1E*. In the rest of the analyses, we use time-averaged responses (2.4–4.8 s after sound change). This range was chosen as the biggest window we could use (to improve SNR) while minimizing contamination from the previous or next sound (indeed, blood volume typically lags neuronal activity by 1.5–2 s). In *Figure 2—figure supplement 1*, we show similar cross-correlation matrices between responses to mixtures and either foregrounds or backgrounds, to confirm that background and foreground invariance are also stable in time.

We computed the voxelwise test–retest correlation by correlating the vector of time-averaged responses to sounds across two repeats, for each category (*Figure 2B*). We then focused on voxels with a test–retest correlation above 0.3 in at least one category (foregrounds, backgrounds, or mixtures) and above 0 in all categories (which ensures that the noise correction can be applied).

## Voxel-wise correlations

For each voxel, we defined background invariance (resp. foreground invariance) as the noise-corrected correlation between mixtures and corresponding foregrounds (resp. backgrounds), across sounds. We excluded the first and last sound snippet of each block, in which either foreground or background was missing in half of the snippet (see design in *Figure 1B*).

We used a noise-corrected version of Pearson correlation to take into account differences in test–retest reliability across voxels. Specifically, the noise-corrected correlation should provide an estimate of the correlation we would measure in the limit of infinite data (*Kell and McDermott, 2019*). The noise-corrected correlation values were obtained using the following equation:

$$r_{noise-corrected}(\boldsymbol{x}, \boldsymbol{y}) = \frac{0.5 * (corr(\boldsymbol{x}_1, \boldsymbol{y}_1) + corr(\boldsymbol{x}_2, \boldsymbol{y}_2))}{\sqrt{corr(\boldsymbol{x}_1, \boldsymbol{x}_2) * corr(\boldsymbol{y}_1, \boldsymbol{y}_2)}} \tag{1}$$

where $\boldsymbol{x}_1$, $\boldsymbol{x}_2$ (resp. $\boldsymbol{y}_1$, $\boldsymbol{y}_2$) are two repetitions of $\boldsymbol{x}$ (resp. $\boldsymbol{y}$).

After this correction, the differences we observed between brain regions were present regardless of voxels' test–retest reliability, or noise level (*Figure 5—figure supplement 1*). Thus, potential differences in vasculature across regions are unlikely to affect our results.

## ROI analyses

We manually defined ROIs by combining anatomical markers and sound responsiveness. We found CBV responses to be organized in distinct patches in auditory cortex (*Figure 2A*). Using anatomical landmarks (*Radtke-Schuller, 2018*), we defined boundaries between the different functional regions from primary (MEG, medial ectosylvian gyrus) to secondary (dPEG, dorsal ectosylvian gyrus) and tertiary (VP, ventral posterior auditory field).

## Map display

For surface views, we averaged values over the depth of the slice. The transparency (alpha value) of each surface bin was determined by the number of voxels included in the average. Thus, white sections in maps correspond to regions in which no reliable voxels were recorded. All maps in the main figures show the left auditory cortex of ferret L. Maps for other ferrets are shown in *Figure 2—figure supplement 2*.

## Cochlear and spectrotemporal modulation energy estimation

Spectrotemporal modulation energy was calculated by measuring the strength of modulations in filtered cochleagrams (which highlight modulations at a particular temporal rate and/or spectral scale) (*Chi et al., 2005*). Specifically, modulation strength was computed as the standard deviation across time of each frequency channel of the filtered cochleagram, using the same 400 ms windows as for fUSI data. The bank of spectrotemporal filters used was the same as in *Landemard et al., 2021* (scales: 0, 0.25, 0.5, 1, 2, 4, 8 cyc/oct; rates: 0.5, 1, 2, 4, 8, 16, 32, 64 Hz). Modulation energy was then averaged in 10 frequency bins (centers: 27, 50, 91, 167, 307, 1032, 3470 Hz).

We then averaged modulation energy using the same temporal window as for fUSI sound-evoked data (2.4–4.8 s after sound change), after shifting it by -0.8 s to account for hemodynamic delays. We thus obtained a feature matrix (frequency * temporal modulation * spectral modulation) for each sound segment.

We fit a linear model using ridge regression to predict voxel responses from sound features. The feature matrix was z-scored as part of the regression. We used sixfold cross-validation: we split the sound set in six folds (keeping blocks of temporally adjacent sounds in the same fold). One fold was left out and the others were used to fit the weights. The lambda hyperparameter was defined using an inner cross-validation loop. The fitted weights were used to predict responses to left-out sounds. We thus built a matrix of cross-validated predicted responses and applied the same analyses as were applied to the true data (*Figure 4*). Prediction accuracy was defined as the Pearson correlation between measured and cross-validated predicted responses (*Figure 4—figure supplement 1*).

To examine the fitted weights, we averaged the weights obtained in all cross-validation folds and multiplied them by the standard deviation of the feature matrix. To extract voxels' preferred value over one feature axis (e.g. frequency), we first averaged the weight matrix along the other feature axes (e.g. spectral and temporal modulation) and used the value of the feature for which the maximum was reached. For plotting the tuning maps and the analysis of voxels tuned to low or high rates, we only considered voxels with average prediction accuracy in left-out folds higher than 0.3.

## Human analyses

To directly compare our results with those obtained in humans, we used the dataset from *Kell and McDermott, 2019*. Specifically, we used data from Experiment 1 which provided the closest match to our experimental conditions, and only considered the last seven subjects that heard both the foregrounds and the backgrounds in isolation, in addition to the mixtures. In brief, subjects were presented with 30 natural foregrounds, 30 natural backgrounds, and mixtures of both. Foregrounds and backgrounds were determined based on a criterion of stationarity, similar to our study. For each subject, backgrounds and foregrounds were matched at random (so that the mixtures were different for each subject). Each sound was presented four times. The SNR (ratio of power of foreground vs. background sound) was -6 dB (vs. 0 dB in our case). To encourage attention to each stimulus, participants performed a sound intensity discrimination task on the stimuli. We used a similar criterion for voxel inclusion and conducted the same analyses as for ferrets. Ridge regression was performed using 10-fold cross-validation. For each subject, (foreground, background, mixture) triplets were kept together in the same fold.

## Statistical testing

To test the difference between background and foreground invariance for each ROI, we ran a permutation test. We built the null distribution by randomly permuting foreground and background labels for each sound and computing invariance for each voxel, 1000 times. We then compared the median difference between background and foreground invariance across voxels, to the values obtained in this null distribution and used the proportion of null values higher than the observed difference to obtain our *p*-value.

To compute statistical significance between ROIs, we performed a permutation test. We repeated the following procedure for each pair of ROI (MEG vs. dPEG, dPEG vs. VP). We built a null distribution by shuffling ROI labels for voxels belonging to the pair of ROIs within each animal, 1000 times. We then compared the observed average across animals of the difference of median correlations between ROIs across animals to this null distribution and used the proportion of null values more extreme than the observed difference to obtain our *p*-value. We also computed a *p*-value for each ferret separately (see *Figure 2—source data 1*). Finally. we ran this test by first focusing on voxels tuned to high or low rates within each ROI.

To test for differences in invariance between voxels tuned to low (< 8 Hz) or high (> 8 Hz) temporal modulation rates within each ROI, we used a similar permutation test in which we compared the difference of invariance between voxels tuned to low vs. high rates to a null distribution in which we shuffled the tuning values across voxels within each animal and ROI.

To compare results between ferrets and humans, we performed Wilcoxon rank-sum tests between ferrets and human subjects for different metrics: median prediction accuracy across voxels, median

noise-corrected correlation between measured and predicted invariance. For the latter, we also tested the difference to zero using a sign-rank test.

## Acknowledgements

We thank Shihab Shamma and Sam V Norman-Haignere for fruitful discussions, Alex JE Kell for providing and helping with the human data, Balkis Cadi and Lynda Bourguignon for technical and administrative support throughout the project. This work was supported by the Institut Universitaire de France, ANR-17-EURE-0017, ANR-10-IDEX-0001–02, as well as an AMX doctoral fellowship to AL.

## Additional information

### Funding

| Funder | Grant reference number | Author |
|---|---|---|
| Agence Nationale de la Recherche | ANR-17- EURE-0017 | Yves Boubenec |
| Agence Nationale de la Recherche | ANR-10-IDEX-0001-02 | Yves Boubenec |
| Institut Universitaire de France | | Yves Boubenec |

The funders had no role in study design, data collection and interpretation, or the decision to submit the work for publication.

### Author contributions

Agnès Landemard, Conceptualization, Formal analysis, Investigation, Methodology, Writing – original draft; Célian Bimbard, Conceptualization, Methodology, Writing – review and editing; Yves Boubenec, Conceptualization, Supervision, Funding acquisition, Methodology, Writing – review and editing

### Author ORCIDs

Agnès Landemard http://orcid.org/0000-0001-6081-1014
Célian Bimbard https://orcid.org/0000-0002-6380-5856
Yves Boubenec https://orcid.org/0000-0002-0106-6947

### Ethics

Experiments were approved by the French Ministry of Agriculture (protocol authorization: 21022) and strictly comply with the European directives on the protection of animals used for scientific purposes (2010/63/EU).

Reviewer #1 (Public review): https://doi.org/10.7554/eLife.106628.3.sa1
Reviewer #2 (Public review): https://doi.org/10.7554/eLife.106628.3.sa2
Reviewer #3 (Public review): https://doi.org/10.7554/eLife.106628.3.sa3
Author response https://doi.org/10.7554/eLife.106628.3.sa4

## Additional files

### Supplementary files

MDAR checklist

### Data availability

Data is publicly available on Zenodo: https://doi.org/10.5281/zenodo.15800440. Code to reproduce the figures and analyses presented in this manuscript is available on Github: https://github.com/agneslandemard/naturalsoundmixtures (copy archived at *Landemard, 2024*).

The following dataset was generated:

| Author(s) | Year | Dataset title | Dataset URL | Database and Identifier |
|---|---|---|---|---|
| Landemard A | 2025 | fUS imaging of ferret auditory cortex during passive listening to natural sound mixtures | https://doi.org/10.5281/zenodo.15800440 | Zenodo, 10.5281/zenodo.15800440 |

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
