## [Editor Report · eLife Assessment]

This paper presents **valuable** findings on the processing of sound mixtures in the auditory cortex of ferrets, a species widely used for studies of auditory processing. Using the convenient and relatively high-resolution method of functional ultrasound imaging, the authors provide **convincing** evidence that background noise invariance emerges across the auditory cortical processing hierarchy. They also draw informative comparisons with previously published fMRI data obtained in humans. This work will be of interest to researchers studying the auditory cortex and the neural mechanisms underlying auditory scene analysis and hearing in noise.

---

## [Referee Report · Reviewer #1 (Public review)]

This is a very interesting paper addressing the hierarchical nature of the mammalian auditory system. The authors use an unconventional technique to assess brain responses -- functional ultrasound imaging (fUSI). This measures blood volume in cortex at a relatively high spatial resolution. They present dynamic and stationary sounds in isolation and together, and show that the effect of the stationary sounds (relative to the dynamic sounds) on blood volume measurements decreases as one ascends the auditory hierarchy. Since the dynamic/stationary nature of sounds is related to their perception as foreground/background sounds, this suggests that neurons in higher levels of the cortex may be increasingly invariant to background sounds.

The study is interesting, well conducted and well written. In the revised manuscript, the authors have addressed all the points I raised in my review.

---

## [Referee Report · Reviewer #2 (Public review)]

Summary:

Noise invariance is an essential computation in sensory systems for stable perception across a wide range of contexts. In this paper, Landemard et al. perform functional ultrasound imaging across primary, secondary and tertiary auditory cortex in ferrets to uncover the mesoscale organization of background invariance in auditory cortex. Consistent with previous work, they find that background invariance increases throughout the cortical hierarchy. Importantly, they find that background invariance is largely explained by progressive changes in spectro-temporal tuning across cortical stations which are biased towards foreground sound features. To test if these results are broadly relevant, they then re-analyze human fMRI data and find that spectro-temporal tuning fails to explain background invariance in human auditory cortex.

Strengths:

(1) Novelty of approach: Though the authors have published on this technique previously, functional ultrasound imaging offers unprecedented temporal and spatial resolution in a species where large-scale calcium imaging is not possible and electrophysiological mapping would take weeks or months. Combining mesoscale imaging with a clever stimulus paradigm, they address a fundamental question in sensory coding.

(2) Quantification and execution: the results are generally clear and well supported by statistical quantification.

(3) Elegance of modeling: The spectrotemporal model presented here is explained clearly and most importantly, provides a compelling framework for understanding differences in background invariance across cortical areas.

Comments on revised version:

The authors have addressed all of my previous concerns and their publicly shared data is easy to view, this is a nice contribution to the field.

---

## [Referee Report · Reviewer #3 (Public review)]

This paper investigates invariance to natural background noise in the auditory cortex of ferrets and humans. The authors first replicate, in ferrets, a finding from human neuroimaging showing that invariance to background noise increases along the cortical hierarchy (i.e. from primary to non-primary auditory cortex). Next, the authors ask whether this pattern of invariance could be explained by differences in tuning to low-level acoustic features across primary and non-primary regions. The authors conclude that this tuning can explain the spatial organization of background invariance in ferrets, but not in humans. The conclusions of the paper are well supported by the data.

The paper is very straightforwardly written, with a generally clear presentation including well-designed and visually appealing figures. Not only does this paper provide an important replication in a non-human animal model commonly used in auditory neuroscience, but also it extends the original findings in three ways. First, the authors reveal a more fine-grained gradient of background invariance by showing that background invariance increases across primary, secondary and tertiary cortical regions. Second, the authors address a potential mechanism that might underlie this pattern of invariance by considering whether differences in tuning to frequency and spectrotemporal modulations across regions could account for the observed pattern of invariance. The spectrotemporal modulation encoding model used here is a well-established approach in auditory neuroscience and seems appropriate for exploring potential mechanisms underlying invariance in auditory cortex, particularly in ferrets. Third, the authors provide a more complete picture of invariance by additionally analyzing foreground invariance, a complementary measure not explored in the original study.

Comments on author revisions:

The authors have thoroughly addressed the concerns raised in my initial review.

---

## [Author Response]

The following is the authors’ response to the original reviews.

**Reviewer #1(Public review):**
(1) Changes in blood volume due to brain activity are indirectly related to neuronal responses. The exact relationship is not clear, however, we do know two things for certain: (a) each measurable unit of blood volume change depends on the response of hundreds or thousands of neurons, and (b) the time course of the volume changes are slow compared to the potential time course of the underlying neuronal responses. Both of these mean that important variability in neuronal responses will be averaged out when measuring blood changes. For example, if two neighbouring neurons have opposite responses to a given stimulus, this will produce opposite changes in blood volume, which will cancel each other out in the blood volume measurement due to (a). This is important in the present study because blood volume changes are implicitly being used as a measure of coding in the underlying neuronal population. The authors need to acknowledge that this is a coarse measure of neuronal responses and that important aspects of neuronal responses may be missing from the blood volume measure.

The reviewer is correct: we do not measure neuronal firing but use blood volume as a proxy for bulk local neuronal activity, which does not capture the richness of single neuron responses. This is why the paper focuses on large-scale spatial representations as well as cross-species comparison. For this latter purpose, fMRI responses are on par with our fUSI data, with both neuroimaging techniques showing the same weakness. We have now added this point to the discussion:

“Second, we used blood volume as a proxy for local neuronal activity. Thus, our signal ignores any heterogeneity that might exist at the level of local neuronal populations. However, our main findings are related to the large-scale organization of cortical responses and how they relate to those of humans. For this purpose, the functional spatial resolution of our signal, driven by the spatial resolution of neurovascular coupling, should be adapted. In addition, using hemodynamic signals provides a much better comparison with human fMRI data, where the same limitations are present.”

(2) More importantly for the present study, however, the effect of (b) is that any rapid changes in the response of a single neuron will be cancelled out by temporal averaging. Imagine a neuron whose response is transient, consisting of rapid excitation followed by rapid inhibition. Temporal averaging of these two responses will tend to cancel out both of them. As a result, blood volume measurements will tend to smooth out any fast, dynamic responses in the underlying neuronal population. In the present study, this temporal averaging is likely to be particularly important because the authors are comparing responses to dynamic (nonstationary) stimuli with responses to more constant stimuli. To a first approximation, neuronal responses to dynamic stimuli are themselves dynamic, and responses to constant stimuli are themselves constant. Therefore, the averaging will mean that the responses to dynamic stimuli are suppressed relative to the real responses in the underlying neurons, whereas the responses to constant stimuli are more veridical. On top of this, temporal following rates tend to decrease as one ascends the auditory hierarchy, meaning that the comparison between dynamic and stationary responses will be differently affected in different brain areas. As a result, the dynamic/stationary balance is expected to change as you ascend the hierarchy, and I would expect this to directly affect the results observed in this study.It is not trivial to extrapolate from what we know about temporal following in the cortex to know exactly what the expected effect would be on the authors' results. As a first-pass control, I would strongly suggest incorporating into the authors' filterbank model a range of realistic temporal following rates (decreasing at higher levels), and spatially and temporally average these responses to get modelled cerebral blood flow measurements. I would want to know whether this model showed similar effects as in Figure 2. From my guess about what this model would show, I think it would not predict the effects shown by the authors in Figure 2. Nevertheless, this is an important issue to address and to provide control for.

We understand the reviewer’s concern about potential differences in response dynamics in stationary vs non-stationary sounds. It seems that the reviewer is concerned that responses to foregrounds may be suppressed in non-primary fields because foregrounds are not stationary, and non-primary regions could struggle to track and respond to these sounds. Nevertheless, we observed the contrary, with non-primary regions overrepresenting non-stationary (dynamic) sounds, over stationary ones. For this reason, we are inclined to think that this explanation cannot falsify our findings.

We understand the comment that temporal following rates might differ across regions in the auditory hierarchy and agree. In fact, we do show that tuning to temporal rates differs across regions and partly explains the differences in background invariance we observe. In this regard, we think the reviewer’s suggestion is already implemented by our spectrotemporal model, which incorporates the full range of realistic temporal following rates (up to 128 Hz). The temporal averaging is done as we take the output of the model (which varies continuously through time) and average it in the same window as we used for fUSI data. When we fit this model to the ferret data, we find that voxels in non-primary regions, especially VP (tertiary auditory cortex), tend to be more tuned to low temporal rates (Figure 2F, G), and that background invariance is stronger in voxels tuned to low rates. This is, however, not true in humans, suggesting that background invariance in humans relies on different computational mechanisms. We have added a sentence to clarify this: “The model included a range of realistic temporal rates and this axis was the most informative to discriminate foregrounds from backgrounds.”

(3) I do not agree with the equivalence that the authors draw between the statistical stationarity of sounds and their classification as foreground or background sounds. It is true that, in a common foreground/background situation - speech against a background of white noise - the foreground is non-stationary and the background is stationary. However, it is easy to come up with examples where this relationship is reversed. For example, a continuous pure tone is perfectly stationary, but will be perceived as a foreground sound if played loudly. Background music may be very non-stationary but still easily ignored as a background sound when listening to overlaid speech. Ultimately, the foreground/background distinction is a perceptual one that is not exclusively determined by physical characteristics of the sounds, and certainly not by a simple measure of stationarity. I understand that the use of foreground/background in the present study increases the likely reach of the paper, but I don't think it is appropriate to use this subjective/imprecise terminology in the results section of the paper.

We appreciate the reviewer’s comment that the classification of our sounds into foregrounds and backgrounds is not verified by any perceptual experiments. We use those terms to be consistent with the literature (McWalter and McDermott, 2018; McWalter and McDermott, 2019), including the paper we derived this definition from (Kell et al., 2019). These terms are widely used in studies where no perceptual or behavioral experiments are included, and even when animals are anesthetized. We have clarified and justified this choice in the beginning of the Results section:

“We used three types of stimuli: foregrounds, backgrounds, and combinations of those. We use those terms to refer to sounds differing in their stationarity, under the assumption that stationary sounds carry less information than non-stationary sounds, and are thus typically ignored.”

We have also added a paragraph in the discussion to emphasize the limits of this definition:

“First, this study defined foregrounds and backgrounds solely based on their acoustic stationarity, rather than perceptual judgments. This choice allowed us to isolate the contribution of acoustic factors in a simplified setting. Within this controlled framework, we show that acoustic features of foreground and background sounds drive their separation in the brain and the hierarchical extraction of foreground sound features.”

(4) Related to the above, I think further caveats need to be acknowledged in the study. We do not know what sounds are perceived as foreground or background sounds by ferrets, or indeed whether they make this distinction reliably to the degree that humans do. Furthermore, the individual sounds used here have not been tested for their foreground/background-ness. Thus, the analysis relies on two logical jumps - first, that the stationarity of these sounds predicts their foreground/background perception in humans, and second, that this perceptual distinction is similar in ferrets and humans. I don't think it is known to what degree these jumps are justified. These issues do not directly affect the results, but I think it is essential to address these issues in the Discussion, because they are potentially major caveats to our understanding of the work.

We agree with the reviewer that the foreground-background distinction might be different in ferrets. In anticipation of that issue, we had enriched the sound set with more ecologically relevant sounds, such as ferret and other animal vocalizations. Nevertheless, we have emphasized this limitation in addition to the limitation of our definition of foregrounds and backgrounds in the discussion:

“In addition, most of the sounds included in our study likely have more relevance for humans compared to ferrets (see table 1). Despite including ferret vocalizations and environmental sounds that are more ecologically relevant for ferrets, it is not clear whether ferrets would behaviorally categorize foregrounds and backgrounds as humans do. Examining how ferrets naturally orient or respond to foreground and background sounds under more ecologically valid conditions, potentially with free exploration or spontaneous listening paradigms, could help address this issue.”

**Reviewer #2(Public review);**
(1) Interpretation of the cerebral blood volume signal: While the results are compelling, more caution should be exercised by the authors in framing their results, given that they are measuring an indirect measure of neural activity, this is the difference between stating "CBV in area MEG was less background invariant than in higher areas" vs. saying "MEG was less background invariant than other areas". Beyond framing, the basic properties of the CBV signal should be better explored:a) Cortical vasculature is highly structured (e.g. Kirst et al.(2020) Cell). One potential explanation for the results is simply differences in vasculature and blood flow between primary and secondary areas of auditory cortex, even if fUS is sensitive to changes in blood flow, changes in capillary beds, etc (Mace et al., 2011) Nat. Methods.. This concern could be addressed by either analyzing spontaneous fluctuations in the CBV signal during silent periods or computing a signal-to-noise ratio of voxels across areas across all sound types. This is especially important given the complex 3D geometry of gyri and sulci in the ferret brain.

We agree with the reviewers that there could be differences in vasculature across subregions of the auditory cortex and note that this point would also be valid for the published human fMRI data. Nevertheless, even if small differences in vasculature were present, it is unlikely that they would affect our analyses and results, which are designed to be independent of local vascular density. First, we normalize the signal in each voxel using the silent periods, so that the absolute strength of the raw signal, or baseline blood volume in each voxel, is factored in our analysis. Second, we only focus on reliably responsive voxels in each region and do see comparable sound-evoked responses in all regions (Figure S2). Third, our analysis mostly relies on voxel-based correlation across sounds, which is independent of the mean and variance of the voxel responses. Differences in noise, measured through test-retest reliability, can affect values of correlation, which is why we used a noise-correction procedure. After this procedure, invariance does not depend on test-retest, and differences across regions are still seen when matching for test-retest (new Figure S7). Thus, we believe that differences in vascular architecture across regions are unlikely to affect our results. We added this point in the Methods section when discussing the noise-correction:

“After this correction, the differences we observed between brain regions were present regardless of voxels' test-retest reliability, or noise level (Figure S7). Thus, potential differences in vasculature across regions are unlikely to affect our results.”

b) Figure 1 leaves the reader uncertain what exactly is being encoded by the CBV signal, as temporal responses to different stimuli look very similar in the examples shown. One possibility is that the CBV is an acoustic change signal. In that case, sounds that are farther apart in acoustic space from previous sounds would elicit larger responses, which is straightforward to test. Another possibility is that the fUS signal reflects time-varying features in the acoustic signal (e.g. the low-frequency envelope). This could be addressed by cross-correlating the stimulus envelope with fUS waveform. The third possibility, which the authors argue, is that the magnitude of the fUS signal encodes the stimulus ID. A better understanding of the justification for only looking at the fUS magnitude in a short time window (2-4.8 s re: stimulus onset) would increase my confidence in the results.

We thank the reviewer for raising that point as it highlights that the layout of Figure 1 is misleading. While Figure 1B shows an example snippet of our sound streams, Figure 1D shows the average timecourse of CBV time-locked to a change in sound (foreground or background, isolated or in a mixture). This is the average across all voxels and sounds, aiming at illustrating the dynamics for the three broad categories. In Figure 1E however, we show the cross-validated cross-correlation of CBV across sounds (and different time lags). To obtain this, we compute for each voxel the response to each sound at each time lag, thus obtaining two vectors (size: number of sounds) per lag, one per repeat. Then, we correlate all these vectors across the two repeats, obtaining one cross-correlation matrix per voxel. We finally average these matrices across all voxels. The presence of red squares with high correlations demonstrates that the signal encodes sound identity, since CBV is more similar across two repeats of the same sound (e.g., in the foreground only matrix, 0-5 s vs 0-5 s), than two different sounds (0-5 s vs. 7-12 s). We modified the figure layout as well as the legend to improve clarity.

(2) Interpretation of the human data: The authors acknowledge in the discussion that there are several differences between fMRI and fUS. The results would be more compelling if they performed a control analysis where they downsampled the Ferret fUS data spatially and temporally to match the resolution of fMRI and demonstrated that their ferret results hold with lower spatiotemporal resolution.

We agree with the reviewer that the use of different techniques might come in the way of cross-species comparison. We already control for the temporal aspect by using the average of stimulus-evoked activity across time (note that due to scanner noise, sounds are presented cut into small pieces in the fMRI experiments). Regarding the spatial aspect, there are several things to consider. First, both species have brains of very different sizes, a factor that is conveniently compensated for by the higher spatial resolution of fUSI compared to fMRI (0.1 vs 2 mm). Downsampling to fMRI resolution would lead to having one voxel per region per slice, which is not feasible. We also summarize results with one value per region, which is a form of downsampling that is fairer across species. Furthermore, we believe that we already established in a previous study (Landemard et al, 2021 eLife) that fUSI and fMRI data are comparable signals. We indeed could predict human fMRI responses to most sounds from ferret fUSI responses to the same identical sounds. We clarified these points in the discussion:

“In addition, fMRI has a worse spatial resolution than fUSI (here, 2 vs. 0.1 mm voxels). However, this difference in resolution compensates for the difference in brain size between humans and ferrets. In our previous work, we showed that a large fraction of cortical responses to natural sounds could be predicted from one species to the other using these methods (Landemard et al., 2021).”

**Reviewer #3 (Public review):**
As mentioned above, interpretation of the invariance analyses using predictions from the spectrotemporal modulation encoding model hinges on the model's ability to accurately predict neural responses. Although Figure S5 suggests the encoding model was generally able to predict voxel responses accurately, the authors note in the introduction that, in human auditory cortex, this kind of tuning can explain responses in primary areas but not in non-primary areas (Norman-Haignere & McDermott, PLOS Biol. 2018). Indeed, the prediction accuracy histograms in Figure S5C suggest a slight difference in the model's ability to predict responses in primary versus non-primary voxels. Additional analyses should be done to (a) determine whether the prediction accuracies are meaningfully different across regions and (b) examine whether controlling for prediction accuracy across regions (i.e., subselecting voxels across regions with matched prediction accuracy) affects the outcomes of the invariance analyses.

The reviewer is correct: the spectrotemporal model tends to perform less well in human non-primary cortex. We believe this does not contradict our results but goes in the same direction: while there is a gradient in invariance in both ferrets and humans, this gradient is predicted by the spectrotemporal model in ferrets, but not in humans (possibly indeed because predictions are less good in human non-primary auditory cortex). Regardless of the mechanism, this result points to a difference across species. In ferrets, we found a significantly better prediction accuracy in VP (p=0.001, permutation test) and no differences between MEG and dPEG (p=0.89). In humans, prediction accuracy was slightly higher in primary compared to non-primary auditory cortex, but this effect was not significant (p=0.076). In both species, when matching prediction accuracy between regions, the gradients in invariance were preserved. We have added these analyses to the manuscript (Figure S5).

A related concern is the procedure used to train the encoding model. From the methods, it appears that the model may have been fit using responses to both isolated and mixture sounds. If so, this raises questions about the interpretability of the invariance analyses. In particular, fitting the model to all stimuli, including mixtures, may inflate the apparent ability of the model to "explain" invariance, since it is effectively trained on the phenomenon it is later evaluated on. Put another way, if a voxel exhibits invariance, and the model is trained to predict the voxel's responses to all types of stimuli (both isolated sounds and mixtures), then the model must also show invariance to the extent it can accurately predict voxel responses, making the result somewhat circular. A more informative approach would be to train the encoding model only on responses to isolated sounds (or even better, a completely independent set of sounds), as this would help clarify whether any observed invariance is emergent from the model (i.e., truly a result of low-level tuning to spectrotemporal features) or simply reflects what it was trained to reproduce.

We thank the reviewer for this suggestion. We have run an additional prediction using only the sounds presented in isolation, which replicates our main results (new Figure S6). We have added this control to the manuscript:

“Results were similar if the model was fit solely on isolated sounds, excluding mixtures from the training set (Figure S6).”

Finally, the interpretation of the foreground invariance results remains somewhat unclear. In ferrets (Figure 2I), the authors report relatively little foreground invariance, whereas in humans (Figure 5G), most participants appear to show relatively high levels of foreground invariance in primary auditory cortex (around 0.6 or greater). However, the paper does not explicitly address these apparent crossspecies differences. Moreover, the findings in ferrets seem at odds with other recent work in ferrets (Hamersky et al. 2025 J. Neurosci.), which shows that background sounds tend to dominate responses to mixtures, suggesting a prevalence of foreground invariance at the neuronal level. Although this comparison comes with the caveat that the methods differ substantially from those used in the current study, given the contrast with the findings of this paper, further discussion would nonetheless be valuable to help contextualize the current findings and clarify how they relate to prior work.

We thank the reviewer for this point. While we found a trend for higher background invariance than foreground invariance in ferret primary auditory cortex, this difference was not significant and many voxels exhibit similar levels of background and foreground invariance (for example in Figure 2D, G). Thus, we do not think our results are inconsistent with Hamersky et al., 2025, though we agree the bias towards background sounds is not as strong in our data. This might indeed reflect differences in methodology, both in the signal that is measured (blood volume vs spikes), and the sound presentation paradigm. Our timescales are much slower and likely reflect responses post-adaptation, which might not be as true for Hamersky et al. We have added this point to the discussion, as well as a comment on the difference between ferrets and humans in foreground invariance in primary auditory cortex:

“In ferrets, primary auditory cortex has been found to over-represent backgrounds in mixtures compared to foregrounds (Hamersky et al., 2025). In contrast, we found a slight, non-significant bias towards foregrounds in primary regions. This difference could be driven by a difference in timescales, as we looked at slower timescales in which adaptation might be more present, reducing the strength of background encoding. In humans, we found a much smaller gap between background and foreground invariance in primary auditory cortex, which was not predicted by the spectrotemporal model. Additional, more closely controlled experiments would be needed to confirm and understand this species difference.”

**Reviewer #1 (Recommendations for the authors):**
(1) In the introduction, explain the relationship between background/foreground and stationarity/non-stationarity, and thus why stationary/nonstationary stimuli could be used to probe differences in background/foreground processing.

We have added a sentence at the beginning of the results section to justify our choice (see public review).

(2) Avoid use of the background/foreground terminology in Results (and probably Methods).

For consistency with previous literature, we decided to keep this terminology, though imperfect. We further justified our choice in the beginning of the Results section (see previous point).

(3) In the Discussion, explain what the implications of the results are for background/foreground processing, and, importantly, highlight any caveats that result from stationarity not being a direct measure of background/foreground.

We added a paragraph in the Discussion to highlight this point choice (see public review).

**Reviewer #2 (Recommendations for the authors):**
(1) Figure 1: Showing a silent period in the examples would help in understanding the fUS signal.

In Figure 1D, we show the average timecourse of CBV time-locked to a change in sound (foreground or background, isolated or in a mixture). This is the average across all voxels and sounds. Thus, it would not be very informative to show an equivalent plot for a silent period, as it would look flat by definition. However, we updated the layout and legend of Figure 1 to make it clearer and avoid confusion.

(2) "Responses were not homogenous" - would make more sense to say something like "responses were not spatially distributed".

We removed these words which were indeed not necessary: “We found that reliable soundevoked responses were confined to the central part of ventral gyrus of the auditory cortex.”

(3) Figure 2D: The maps shown in Figure 2D are difficult to understand for the noninitiated in fUS. At a minimum, labels should be added to indicate A-P, M-L, D-V. I cannot see the white square in the primary figure. An additional graphic would be helpful here to understand the geometry of the measurement.

We thank the reviewer for pointing out that reading these images is indeed an acquired skill. We added an annotated image of anatomy with indications of main features to guide the reader in Figure 1. We also added missing white squares.

(4) Figure 2F: Can the authors better justify why the summary statistic is shown for all three areas, but the individual data only compares primary vs. higher order?`

We now show individual data for all three areas.

(5) More methods information is needed to understand how recordings were stitched across days. Was any statistical modeling used to factor out the influence of day on overall response levels?

We simply concatenated voxels recorded across different sessions and days. The slices were sampled randomly to avoid any systematic effect. Because different slices were sampled in different sessions, any spatial structure spanning several slices is unlikely to be artefactual. For instance, the map of average responses in Figure 2A shows a high level of continuity of spatial patterns across slices. This indicates that this pattern reflects a true underlying organization rather than session-specific noise. It also shows that the overall response levels are not affected by the day or recording session. We added a section in the Methods (“Combining different recordings”) to clarify this point:

“The whole dataset consisted of multiple slices, each recorded in a different recording session. Slices to image on a given day were chosen at random to avoid any systematic bias. Responses were consistent across neighboring slices recorded on different sessions, as shown by the maps of average responses (Figure 2A, Figure S2) where any spatial continuity across different slices must reflect a true underlying signal in the absence of common noise.”

**Reviewer #3 (Recommendations for the authors):**
(1) Figures:The figures are generally very well done and visually appealing. However, I have a few suggestions and questions.a) In Figure 1G, the delta CBV ranges from 0.5 to 1.5, although in subsequent figures (e.g., Figure 2D), the range is much larger (-15 to 45). Is it possible that the first figure is a proportion rather than a percentage, or is there some other explanation for the massive difference in scale? Not being very familiar with this measure, it was confusing.

The same scale is used in both figures, the major difference being that in Figure 1D, we take the average over all voxels and sounds (for each category), which will include many nonresponsive voxels, and for responsive voxels, sounds that they do not respond a lot to. On the other hand, Figure 2D shows the response of a single, responsive voxel. Thus, the values it reaches for its preferred sounds (45%) are an extreme, which weighs only little in Figure 1D. We have changed the legend of Figure 1D to make this more explicit.

b) Similar to the first point, the strength of the correlations in the matrices of Figure 1E is very small (~ 0.05) compared to the test-retest reliabilities plotted in Figure 2B (~0.5). Again, I was confused by this large difference in scale.

Two main factors explain the difference in values between Figure 1E and Figure 2B. First, in Figure 1B, each correlation is done on the average activity in a window of 0.3 s, opposed to 2.4 s in Figure 2B. More averaging leads to better SNR, which inevitably leads to higher testretest correlations. Second, in Figure 1B, the cross-correlation matrices are averaged across all responsive voxels without any criterion for reliability. On the other hand, Figure 2B show example voxels with good test-retest reliability.

c) In Figure 2D, the example voxels are supposed to be shown in white. It appears that this example voxel is only shown for the non-primary voxel. Please be sure to add these voxels throughout the other panels and figures as well.

We fixed this mistake and added the example voxel in all panels.

d) Why do the invariance results (e.g., Figure 2F) for individual animals combine across dPEG and VP, while the overall results (across all animals) split things across all three regions? The results in Table 2 do, in fact, provide this data. Upon further examination of the data in Table 2, it seems like there is only a significant difference between background invariance between dPEG and VP for one of the two animals, and that this might be what drives the effect when pooling across all animals. This seems important to both show visually in the figure and to potentially discuss. There is still very clearly a difference between primary and non-primary, but whether there is a real difference between dPEG and VP seems more unclear.

We added the values for single animals in the plot and highlighted this limitation in the text:

“While background invariance was overall highest in VP, the differences within non-primary areas were more variable across animals (see table 2).”

e) Again, as in Figure 2F, the cross symbols seem like a bad choice as markers since the vertical components of the cross are suggestive of the error of the measurement. However, no error is actually plotted in these figures. I recommend using a different marker and including some measure of error in the invariance plots.

We replaced the crosses with circles to avoid confusion. The measure of error is provided by the representation of values for single animals.

f) The caption for Figure 4C states that each line corresponds to one animal, but does not precisely state what this line represents. Is this the median or something?

Each line indeed represents the median across voxels for one animal. We added this information to the legend.

g) In Figure 5, the captions for panels D and E are swapped.

This has now been corrected.

(2) Discussion:(a) In the paragraph on methodological differences, it mentions that the fMRI voxel size is around 2 mm. This may be true in general, but given the comparison to Kell & McDermott 2019, the voxel size should reflect that used in their study (1 mm).

The reviewer might refer to this sentence from the methods of Kell et al., 2019: “T1weighted anatomical images were collected in each participant (1-mm isotropic voxels) for alignment and cortical surface reconstruction.” However, this does not correspond to the resolution of the functional data, which is 2 mm, as mentioned a bit further in the Methods: “In-plane resolution was 2 × 2 mm (96 × 96 matrix), and slice thickness was 2.8 mm with a 10% gap, yielding an effective voxel size of 2 × 2 × 3.08 mm.”

(b) In the next paragraph on the control of attention, it mentions that attentional differences could play a role. However, in Kell & McDermott 2019, they manipulated attention (attend visual versus attend auditory) and found that it did not substantially affect the observed pattern invariance. I suppose it could potentially affect the degree to which an encoding model could explain the invariance. This seems important, and given that the data was already collected, it could be worth it to analyze that data.

As the reviewer points out, Kell et al. 2019 ran an additional experiment in which they manipulated auditory vs. visual attention. However, the auditory task was just based on loudness and ensured that the participants were awake and paying attention to the stimuli, but not specifically to the foreground or background. This type of attention did not lead to changes in the observed patterns of invariance, which might have been the case for selective attention to backgrounds or foregrounds in the mixture. Given that these manipulations were not done in the ferret experiments, we chose to not include the analysis of this dataset in the scope of this paper. However, future work investigating that topic further would indeed be of interest.

(c) The mention of "a convolutional neural network trained to recognize digits in noise" should make more obvious that this is visual recognition rather than auditory recognition.

We clarified this sentence to make clear that the recognition is visual and not auditory: “For instance, in a convolutional neural network trained to visually recognize digits in different types of noise, when local feedback is implemented, early layers encode noise properties, while later layers represent clean signal.”

(d) Finally, one explanation of the results in the discussion is that "primary auditory areas could be recruited to maintain background representations, enabling downstream cortical regions to use these representations to specifically suppress background information and enhance foreground representations." This "background-related information" being used to "facilitate further extraction of foregrounds" is similar to what is argued in Hicks & McDermott PNAS 2024.

We thank the reviewer for suggesting this relevant reference and added it in this paragraph of the discussion.

(3) Methods:In the "Cross-correlation matrices" section, it mentions that time-averaged responses from 2.4 to 4.8 s were used. It would be helpful to provide an explanation of why this particular time window was used. Additionally, I wondered whether one could look at adaptation type effects (e.g., that of Khalighinejad et al., 2019) or whether fUSI does not offer this kind of temporal precision?

The effects shown in Khalighinejad et al., 2019, are indeed likely too fast to be observed with our methods. However, there are still dynamics in the fUSI signal and in its invariance (Figure S1). Each individual combination of foreground and background is presented for 4.8 s (Figure 1B). Therefore, we chose the range 2.4-4.8 s as the biggest window we could use (to improve SNR) while minimizing contamination from the previous or next sound (indeed, blood volume typically lags neuronal activity by 1.5-2 s). We added this precision to the methods.

In the "Human analyses" section, it is very unclear which set of data was used from Kell & McDermott 2019. For example, that paper contains 4 different experiments, none of which has 7 subjects. Upon closer reading, it seems that only 7 of the 11 participants from Experiment 1 also heard the background sounds in isolation (thus enabling the foreground invariance analyses). However, they stated that there were only 3 female participants in that experiment, while you state that you used data from 7 females. It would be helpful to double-check this and to more clearly state exactly which participants (i.e., from which experiment) were used and why (e.g., why not use data from Experiment 4 in the visual task/attention condition?).

We added a sentence to clarify which datasets were used: “Specifically, we used data from Experiment 1 which provided the closest match to our experimental conditions, and only considered the last 7 subjects that heard both the foregrounds and the backgrounds in isolation, in addition to the mixtures.”

It was a mistake to mention that it was all female, as the original dataset has 3 females and 8 males, of which we used 7 without any indication of their sex. Thus, we removed this mention from the text.

In the "Statistical testing" section, why were some tests done with 1000 permutations/shuffles while others were done with 2000?

We homogenized and used 1000 permutations/shuffles for all statistical tests.

(4) Miscellany:(a) The Hamersky et al. 2023 preprint has recently been published (referenced in the public review), and so you could consider updating the reference.

This reference has now been updated.

(b) There are a few borderline statistical tests that could use a bit more nuance. For example (on page 4), "In primary auditory cortex (MEG), there was no significant difference between values of foreground invariance and background invariance (p = 0.063, obtained by randomly permuting the sounds' background and foreground labels, 1000 times)." This test is quite close to being significant, and this might be acknowledged.

We emphasized the trend to nuance the interpretation of these results: “In primary auditory cortex (MEG), foreground invariance was slightly lower than background invariance, although this difference was not significant (p=0.063, obtained by randomly permuting the sounds' background and foreground labels, 1000 times).”

(5) Potential typos:(a) Should the title be "natural sound mixtures" instead of "natural sounds mixtures"?(b) The caption for Figure 1 says "We imaged the whole auditory through successive slices across several days." I believe this should the "the whole auditory [cortex]." (c) In the first paragraph of the discussion, there is a sentence ending in "...are segregated in hemody-namic signal." I believe this should be "hemody-namic signal."

These errors are now all corrected.